# Inflammatory Processes: Key Mediators of Oncogenesis and Progression in Pancreatic Ductal Adenocarcinoma (PDAC)

**DOI:** 10.3390/ijms252010991

**Published:** 2024-10-12

**Authors:** Liu Yang, Shuangying Qiao, Ge Zhang, Aiping Lu, Fangfei Li

**Affiliations:** 1Shum Yiu Foon Shum Bik Chuen Memorial Centre for Cancer and Inflammation Research, School of Chinese Medicine, Hong Kong Baptist University, Hong Kong SAR, China; yangliu_rachel@life.hkbu.edu.hk (L.Y.); qiaosysy@hkbu.edu.hk (S.Q.); 2Institute of Precision Medicine and Innovative Drug Discovery (PMID), School of Chinese Medicine, Hong Kong Baptist University, Hong Kong SAR, China; zhangge@hkbu.edu.hk; 3Law Sau Fai Institute for Advancing Translational Medicine in Bone & Joint Diseases, School of Chinese Medicine, Hong Kong Baptist University, Hong Kong SAR, China

**Keywords:** pancreatic ductal adenocarcinoma, inflammation, immunity, tumor microenvironment, metabolism, microbiome

## Abstract

Associations between inflammation and cancer were first discovered approximately 160 years ago by Rudolf Virchow, who observed that tumors were infiltrated with inflammatory cells, and defined inflammation as a pathological condition. Inflammation has now emerged as one of the key mediators in oncogenesis and tumor progression, including pancreatic ductal adenocarcinoma (PDAC). However, the role of inflammatory processes in cancers is complicated and controversial, and the detailed regulatory mechanisms are still unclear. This review elucidates the dynamic interplay between inflammation and immune regulation, microenvironment alteration, metabolic reprogramming, and microbiome risk factors in PDAC, committing to exploring a deeper understanding of the role of crucial inflammatory pathways and molecules for providing insights into therapeutic strategies.

## 1. Background

As an essential part of the digestive system, the pancreas is situated in the upper abdomen close to the small intestine and directly behind the stomach. It is composed of two separate parts: the endocrine islets, which produce a variety of hormones, most of which are involved in controlling blood sugar, such as insulin, glucagon, somatostatin, and pancreatic polypeptide, and the exocrine pancreas, which secretes digestive enzymes that aid in food digestion. Amylase, for example, breaks down carbohydrates into glucose; proteases, proteins into amino acids; and lipase, fats into fatty acids and glycerol. A properly working pancreas is essential to a healthy digestive system. When pancreatic dysfunction occurs, the body may not be able to maintain stable blood sugar levels or digest food properly.

In addition, pancreatic cancer can be divided into two main categories: exocrine pancreatic cancer accounting for 95%, and neuroendocrine pancreatic cancer, depending on the tissue origin and pathological type. PDAC, the exocrine pancreatic cancer and the most common pathological type of pancreatic cancer, begins in the cells lining the ducts that carry digestive enzymes out of the pancreas [1,2,3,4]. Because of the prevalence of this type of pancreatic cancer, we focus on PDAC when discussing the relationship between pancreatic cancer and inflammation in this review.

According to the Global Cancer Statistics in 2020 [5], per year, pancreatic cancer (PC), particularly PDAC, is a highly devastating malignancy and its mortality (466,000) closely parallels the incidence (496,000) globally with a 5-year survival rate of less than 6% in both sexes characterized by its late-stage diagnosis, invasiveness, rapid progression, metastatic potential, lack of effective therapeutic strategies, resistance to conventional therapy, and dismal prognosis [1,3,6,7,8,9,10,11,12,13,14,15,16,17,18]. PDAC is the fourth leading cause of cancer deaths in the Western world. Given that the rates of this disease are rather stable relative to the declining rates of breast cancer, it was projected in a study of 28 European countries that PDAC will surpass breast cancer as the third leading cause of cancer death by 2025 [5,19]. Meanwhile, the prediction curves predict that it will be the second leading cause of cancer-related death around 2030, just after lung cancer.

In recent years, emerging evidence demonstrates that inflammation is strongly associated with pancreatic cancer, potentially representing key mediators [3,4]. In one mechanism, inflammation promotes the survival and growth of cancer cells through the production of inflammatory mediators such as cytokines. In another way, inflammation is a mediator of immunosurveillance suppression. In the broadest sense, inflammatory cells might affect and be affected by tumor cell metabolism. Considering all the above-mentioned factors, new discoveries and therapeutic avenues for PDAC have arisen simultaneously, particularly in the context of immunotherapies or anti-inflammatory drugs [20,21,22].

Next, we focus on the dynamic and evolving interplay between inflammation with immune regulation, microenvironment alteration, metabolic reprogramming, and microbiome risk factors for PDAC, targeting various immune cells, tumor cells, inflammatory cells, mutation, chemokines, and cytokines in PDAC (Figure 1). We also summarize the current research on inflammatory pathways and molecules that are expected to become effective therapeutic strategies for PDAC.

## 2. Inflammation and Immune Infiltrations in the PDAC Tumor Microenvironment

Recent studies have demonstrated that the pancreatic cancer microenvironment plays a critical role in enhancing tumor initiation, growth, metastasis, therapy resistance, and disease recurrence development. Inflammation is particularly important in PDAC compared to other cancer types for various reasons: PDAC is frequently linked to chronic pancreatitis; PDAC is distinguished by a dense fibrotic stroma, known as a desmoplastic response, which is triggered by inflammatory cells and cytokines; inflammatory processes can induce DNA damage and mutations, contributing to the genetic instability that is characteristic of PDAC; and PDAC is renowned for its ability to escape the immune system. The inflammatory environment can promote the recruitment of immune-suppressive cells, preventing effective anti-tumor immunity. These factors collectively contribute to the aggressive nature of PDAC, making it distinct from other cancer types.

Tumor cells, which induce substantial cellular, molecular, and physiological adaptations, as well as stromal cells and surrounding extracellular matrix (ECM) components, establish the PDAC tumor microenvironment (TME) [2,12,23,24]. PDAC tissue is infiltrated by a wide variety of immune cells, dominated by immunosuppressive cells, such as regulatory T cells (Tregs), myeloid-derived suppressor cells (MDSCs), and tumor-associated macrophages (TAMs) [7,12,17,23,25,26,27,28]. These immunosuppressive cells and other varieties of cell types including cancer-associated fibroblasts (CAFs) and pancreatic stellate cells (PSCs) are a vital ingredient of the PDAC TME and can also release cytokines and chemokines to indirectly alter the PDAC TME, such as interleukin 1β (IL-1β), IL-6, IL-18, tumor necrosis factor alpha (TNFα), IL-10, and transforming growth factor beta (TGFβ) [2,12,15,23,24,29,30,31]. Additionally, due to the interaction of those cells in the PDAC TME, three hallmarks—immunosuppression, stromal desmoplasia, and hypoxic areas—appear and they are strongly associated with the prognosis of PDAC (Table 1).

### 2.1. Immunosuppressive Cells in PDAC-Related Inflammation

#### 2.1.1. Regulatory T Cells (Tregs)

Tregs are one of the major immunosuppressive cell subsets in the pancreatic TME [2,23]. Tumor cells can escape host immunity by producing immunosuppressive cytokines as well as by recruiting regulatory immune cells with immunosuppressive functions, such as Treg and MDSCs, which in turn can produce immunosuppressive cytokines [3,32]. Higher proportions of Tregs are associated with progression and a poorer prognosis in patients with PDAC [18,24].

Tregs influence tumor growth by acting either on supporting the survival of cancer cells through the secretion of growth factors or via the inhibition of immune cell effector functions through multiple mechanisms, including the expression of IL-10, TGF-β, CTLA-4, and granzyme B. On the one hand, cancer cell-derived IL-1α induces the recruitment of Tregs to foster the formation of an immunosuppressive situation by increasing CCL22 expression. PDAC cancer cells secrete nuclear factor kappa-B (NF-κB) [12], TGF-β, and CCL5, a critical modulator of inflammation and an important chemoattractant in recruiting Treg into tumors. Similarly, TNFα produced by cancer and inflammatory cells within the TME can also recruit Tregs and impair immune surveillance by suppressing T-cell responses and the cytotoxic activity of activated macrophages. On the other hand, Tregs secrete IL-10 and TGFβ to inhibit effector T-cell functions and express granzyme B to induce effector T-cell cytolysis. Tregs also can downgrade the activity of CD4+, CD8+, and Natural killer cells (NK cells). In addition, Tregs can bind to IL-2 competitively to inhibit effector cells [12,17,23,32,33,34]. There is a close association between the expression of CXCL10, which is a chemokine expressed in many inflammatory diseases with the presence of Tregs, and an immunosuppressed microenvironment [29]. Foxp3+ Tregs constitute a subtype of T cells with significant roles in immunosuppression modulation during PDAC progression [9,12,28,31].

#### 2.1.2. Myeloid-Derived Suppressor Cells (MDSCs)

MDSCs are a heterogeneous group of cells from the myeloid lineage that expands under pathologic conditions including cancer and inflammation and that can inhibit lymphocyte function and suppress T-cell responses [12,32,34]. Like Tregs, MDSCs display an immunosuppressive phenotype, dominate the early immune response, and persist as PDAC becomes invasive, yet there are few antitumor effector T cells [4]. Cancer cells can escape host immunity by recruiting MDSCs, which in turn can produce cytokines with immunosuppressive functions [3,32]. For example, lactate, the main metabolite of aerobic glycolysis in tumor cells, can induce MDSC expansion, which tends to be immunosuppressed. Importantly, the hypoxic feature of PDAC TME is one of the key factors for MDSC recruitment.

MDSCs can suppress tumor immunity through a variety of pathways. For instance, MDSCs secrete IL-10, TGF-β, indoleamine 2,3-dioxygenase 1 (IDO1), and reactive oxygen species (ROS), inducing an increase in Tregs activation, inhibiting the function of NK cells, causing oxidative stress in T cells, and indirectly inducing immunosuppression [3,23,29]. Lesina, M., et al. [27] explained in 2011 that IL-6 released by myeloid cells induces transcription 3 (STAT3) activation in the pancreas, thus potentiating a feed-forward response in the TME to promote pancreatic intraepithelial neoplasia (PanIN) progression and PDAC development by using the Kras^G12D^ model in the pancreas. A recent study showed that blocking IL-8 decreased MDSC recruitment and improved the anti-tumor effectiveness of NK cells. IL-1 and IL-17 also can facilitate tumor growth by promoting the formation of an immunosuppressive microenvironment and inducing angiogenesis.

#### 2.1.3. Tumor-Associated Macrophages (TAMs)

Depending on their phenotype, macrophages can either promote or inhibit tumor progression. [35] In this section, we focus on TAMs, another type of immune suppressor cell. TAMs mainly derive from monocytes in peripheral blood. TAMs appear to be the major source of pro-inflammatory IL-6 in murine and human PDAC. Several studies have provided strong evidence that IL-6 induces STAT3 activation in the pancreas via IL-6 trans-signaling, leading to disease progression [27,28,35,36]. Coincidentally, Gregory L., et al. [35] explored the efficacy of CD40 Agonists in PDAC, which was published in Science, and also produced the idea that TAMs secrete IL-6, which directly impacts the epithelium in PDAC. Like MDSCs, TAMs also can secrete IL-10, IL-17, and indoleamine 2,3-dioxygenase 1 (IDO1), which enhances Treg activation and expansion. Then, Tregs mediate the inhibition of antitumor activity, increase tumor growth, and reduce the survival rate [3,23,29]. By secreting several pro-angiogenic substances, the vascular endothelial growth factor (VEGF), and IL-35, TAMs can stimulate angiogenesis to increase the tumor’s blood supply and promote metastasis.

Ye, H., et al. [37] confirmed that CCL18 secreted by TAMs works via binding to PITPNM3 and induces VCAM-1 expression mostly through the activation of the NF-κB pathway in PDAC cells. In their most recent work, Xia, Q., et al. [38] investigated the link between programmed death-ligand 1 (PD-L1), pyruvate kinase M2 (PKM2), and poor prognosis of PDAC and discovered that TGF-β1 released by TAMs upregulates PD-L1 levels in PDAC cells by stimulating dimeric PKM2 to translocate into the nucleus. Additionally, it was discovered that blocking the TGF-β1-mediated PKM2/STAT1-PD-L1 axis could change the immune microenvironment by activating NK cells and preventing tumor growth.

### 2.2. Other Cell Types in PDAC-Related Inflammation

#### 2.2.1. Cancer-Associated Fibroblasts (CAFs)

CAFs are derived from a variety of cell types, including tissue-resident fibroblasts, quiescent stellate cells, mesenchymal stem cells (MSCs), adipocytes, pericytes, smooth muscle cells, endothelial cells, epithelial cells, and monocytes via various mechanisms. During the development of a tumor, immune and cancer cells can both stimulate CAFs. Immune suppression can potentially be mediated by activated CAFs in PDAC TME. Erez, N., et al. [39] discovered that IL-1β secreted from resident immune cells and tumor cells can reprogram normal fibroblasts into pro-inflammatory CAFs in research that was published in Cancer Cell in 2011. These activated CAFs further mediate tumor-enhancing inflammation by recruiting TAMs and promoting angiogenesis. They also investigated potential cellular sources of IL-1β in the preneoplastic stages and found that fibroblasts, macrophages, and dysplastic epithelial cells can all express IL-1β mRNA in dysplastic tissue using real-time PCR. IL-1β is a pro-inflammatory mediator that is frequently upregulated in a variety of cancers, and its production is associated with poor prognosis. Jiang, H., et al. [28] and Helms, E., et al. [36] illustrated that CAF-derived IL-6 can promote the differentiation of monocyte precursors into MDSCs via STAT3 and promote immune suppression. CAFs can impair cytotoxic T lymphocyte (CTL)-driven antitumor immunity through the production of soluble factors such as IL10, TGF-β, or VEGF. CAF production of CCL17, IL-15, and TGF-β also can promote Treg recruitment and differentiation [28].

#### 2.2.2. Pancreatic Stellate Cells (PSCs)

PSCs are a predominant cell type in the PDAC TME and are significant mediators of the desmoplastic response. Leinwand, J., et al. [17] and Das, S., et al. [40] found that tumor cell-derived IL-1β, regulated by TLR4 and the pancreatic microbiome, promotes the activation and secretory phenotype of quiescent PSCs and establishes desmoplasia mediated by MDSCs and TAMs to establish an immunosuppressive milieu and promote pancreatic tumorigenesis. Similarly, cancer cells can promote the activation of PSCs by producing mitogenic and fibrogenic factors [28]. Hypoxia can induce PSC activation, and then activated PSCs cultured under hypoxia exploit their hypoxia-driven oxidative stress to secrete soluble factors, including IL-6, VEGF-A, and stromal cell-derived factor-1 (SDF-1), favoring angiogenic and inflammatory responses and invasion during PDAC progression [16]. Notably, PSCs may affect tumor metabolism through the production of alanine as an alternative carbon source [41]. In addition, PSCs become more fibrogenic in the presence of hypoxia.

### 2.3. Hallmarks of PDAC Tumor Microenvironment

#### 2.3.1. Immunosuppression and Immune Escape

In the PDAC TME, the inflammatory cell infiltrate is unbalanced toward an immunosuppressive phenotype. Inflammation, caused by the release of pro-inflammatory cytokines and activation of the immune system, might be the accelerator and ultimate factor contributing to the development of PDAC [7]. During cancer evolution, inflammatory cells might orchestrate the escape of tumor cells from immune control by modifying their antigenic fingerprint and favoring a shift toward a more immunosuppressive phenotype. In turn, the epigenetic reprogramming of innate immune cells induces a long-lasting pro-inflammatory phenotype, a phenomenon known as ‘innate immune memory’ or ‘trained immunity’.

#### 2.3.2. Stromal Desmoplasia

A typical feature of the PDAC TME is abundant desmoplasia, which comprises ECM components, fibroblasts, PSCs, and vascular and immune cells [3,4,12]. PSCs are a predominant cell type in the pancreatic tumor stroma and are important mediators of the desmoplastic response [41]. Generally, PSCs activated by inflammatory signals, CAFs, together with collagen deposits promote the desmoplasia of PDAC by secreting certain molecules including the dual-face cytokine TGFβ, fibroblast growth factor 2 (FGF2), and the connective tissue growth factor (CTGF) [24,30,33,40]. There is a feed loop in which CTGF also can directly bind to a5β1 integrin to increase the cell adhesion and migration, in turn altering the activation of PSCs [24]. TAMs also contribute to desmoplasia by facilitating PSCs. Hypoxia can lead to the recruitment of macrophages to activate PSCs through CCL2 secretion induced by HIF1, enhancing desmoplasia by PSCs [28]. Furthermore, flourishing evidence demonstrates that inflammatory cytokines contribute to this process, such as mitogen-activated protein kinase (MAPK) or albumin [24,30]. Indeed, PDAC is a tough tumor, leading to the protection of cancer cells from exogenous anticancer drugs and immune surveillance [24,30,33,40].

#### 2.3.3. Hypoxic Areas

Furthermore, desmoplastic fibrotic stroma and the rapid proliferation of cancer cells can establish a highly hypoxic microenvironment by enhancing the functions of antiangiogenic factors and impairing the vasculature, which increases oxygen consumption and compromises oxygen supply [2,16,23,33,41]. Hypoxia correlates with a state of oxidative stress, and ROS induction is one of the most common regulatory mechanisms under hypoxic conditions [16]. Measuring VEGF and IL-6 under hypoxic conditions, the results showed that their upregulation enhanced the invasive growth of pancreatic cancer cells [33]. Adaptation to the hypoxic condition itself increases the malignant potential of pancreatic cancer cells. Hypoxia has been proposed as a key mediator in the recruitment of MDSCs in the PDAC TME [12].

Commonly, within the restricted TME ecosystem, tumor cells might directly shape immune cells toward an immunosuppressive phenotype through metabolic competition and the direct or exosome-mediated transfer of metabolites, enzymes, and nucleic acids [12,42].

## 3. Inflammation and Metabolism Interplay in PDAC Oncogenesis and Progression

In the broadest sense, inflammatory cells might affect and be affected by tumor cell metabolism. Importantly, inflammation and reprogrammed metabolism may be more crucially involved in PDAC than previously assumed, according to emerging research in recent years [12,43]. PDAC is characterized by a robust fibro-inflammatory response, more fibrosis, and fewer blood vessels. Factually, the PDAC TME is mostly formed by inflammatory cells and a rich fibrotic stroma populated primarily by non-neoplastic fibroblasts and vascular cells. Upon stimulation, fibroblasts deposit many ECM proteins, including the fluid-rich glycosaminoglycan hyaluronan, which is a key factor in increased interstitial pressure [44,45]. This extreme pressure causes tumor hypoperfusion and vascular collapse, which reduces the availability of oxygen and nutrients. Consequently, the metabolism of pancreatic cancer faces significant difficulties in maintaining redox due to a lack of oxygen and nutrients [46]. In this section, we focus on researching the energy metabolism of carbohydrates, proteins, and lipids (Figure 2).

Additionally, much of what has been described for this reprogramming is driven by mutations in the oncogene KRAS, which is nearly universally mutated in PDAC, and inactivating mutations in suppressor genes like TP53, SMAD4, and CDKN2A are thought to be responsible for the growth of pancreatic cancer and its poor prognosis [13,14,15,16,23,26,31,46].

### 3.1. Carbohydrate Metabolism and PDAC-Associated Inflammation

The human body strictly manages its blood glucose levels to ensure appropriate physiological function. An extremely complex network of hormones and neuropeptides, mostly from the brain, pancreas, liver, gut, adipose, and muscle tissue, is responsible for maintaining metabolism [47]. It is worth noting that the pancreas, which consists of endocrine and exocrine glands, is a critical player in this network by secreting the blood sugar-lowering hormone insulin and its antagonist glucagon and digestive enzymes. Cancer cells can reprogram their glucose metabolism, which is characterized by aerobic glycolysis, as first observed by Otto Warburg [12,37,48,49]. Notably, pancreatic cancer cells exhibit extensive glucose metabolic reprogramming, including glycolytic enzyme overexpression and increased lactate production [26,31].

Recent studies have suggested that the expression of glycolytic enzymes is increased in PDAC tissues [16]. TNFα can activate two key regulatory glycolysis enzymes, namely phosphofructokinase and fructose-1.6-bisposphatase [12,31]. Glycolysis promotes the progression of PDAC and reduces cancer cell sensitivity to gemcitabine. A substantial number of studies verified that macrophages secrete a molecule, TNFα, that induces insulin resistance in adipocytes, that obesity was associated with the increased expression of inflammatory mediators in adipose tissue, and that this inflammation interfered with glucose metabolism [12,31,50,51].

### 3.2. Proteins Metabolism and PDAC-Associated Inflammation

Protein metabolism is essential to support PDAC progression. Tumor-associated cells exploit amino acids as substrates to produce molecules endowed with immunomodulatory activities. Glutamine (Gln) is the most abundant in circulation and is a major source of carbon and nitrogen for cancer cells among the amino acids [1,16,26,31,52]. According to research that was recently published in Nature, PDAC is clearly dependent on de novo ornithine synthesis from Gln, which is accomplished by the enzyme ornithine aminotransferase (OAT) and supports the production of polyamines [52]. In the mitochondria, Gln can be metabolized to glutamate (Glu) by glutaminase (GLS). Moreover, Glu and aspartate play critical roles via the tricarboxylic acid (TCA) cycle [14,31,53]. KRAS-driven cancer cells metabolize Gln differently than through the canonical glutamate dehydrogenase (GDH) pathway to produce a-KG. Aspartate generated by mitochondrial glutamate-oxaloacetate transaminase (GOT) exits the mitochondria where it is converted to oxaloacetate by cytosolic GOT and maintains cellular redox homeostasis. This further illustrates aspartate as being critical for cell growth and proliferation [16,53].

Gln also contributes to the cell’s antioxidant defenses through its metabolism to Glu, which is used to produce glutathione (GSH) by stimulating the uptake of cystine. Consequently, the impact of Gln on redox balance involves both the generation of aspartate in the transamination reaction between a-KG and oxaloacetate and the generation of Glu to import cystine for incorporation into GSH [16,53]. Cristovão M. Sousa, et al. [41] and Helms, E., et al. [36] demonstrated that PSCs and CAFs are critical for PDAC metabolism through the secretion of alanine, which preferentially acts as an alternative carbon source in the TCA cycle [36,41]. In addition to Gln, other amino acids are also key players in PDAC progression. PDAC also has an extraordinary amino acid degradation ability, which is critical for cancer development as well [26].

### 3.3. Lipid Metabolism and PDAC-Associated Inflammation

It has been extensively reported that the dysregulation of lipid metabolism is associated with the progression of various cancers, including PDAC, primarily because lipids sustain membrane biosynthesis during rapid proliferation, produce signaling molecules for many cellular activities, and are used in energy storage under conditions of metabolic stress [26,31,54]. In mice models, a high-fat diet can increase oncogenic KRAS activity, leading to fibrosis, inflammation, and enhanced PDAC development [31,55]. In addition, high Ras activity can generate inflammatory mediators via the activation of several mechanisms, including NF-κB, cyclooxygenase 2 (COX2), and others. Consequently, the activation of oncogenic Kras beyond a threshold initiates a Ras-inflammation positive feed-forward loop in which the elevated expression of inflammatory mediators activates and prolongs Kras activity [55]. Meanwhile, inflammatory mediators released from acinar cells under the influence of high Ras activity induce pancreatic fibrosis.

The core molecular mechanism of ferroptosis is involved in the production of lipid peroxidation and subsequent plasma membrane damage [1,56]. Ferroptotic damage can release damage-associated molecular pattern molecules (DAMPs), thereby creating an inflammatory TME for tumor growth [56]. Thus, the inflammatory process has become a key mediator of the development and progression of PDAC. Conversely, it is important to note that many lipids play anti-inflammatory roles and produce metabolic benefits that have been identified [43]. Another novel research study showed that cuproptosis, which is distinct from known death mechanisms, occurs via the direct binding of copper to lipoylated components of the TCA cycle [57,58]. Dating back to the year 2013, Seiko et al. had proven that bioavailable copper was able to modulate oxidative phosphorylation and the growth of tumor tissues [58,59].

### 3.4. Cancer Cachexia Syndrome and PDAC-Associated Inflammation

Cachexia is a major characteristic of advanced and metastatic cancers, and it is highly prevalent in PDAC, affecting almost 70–80%. It is a multifactorial syndrome characterized by non-volitional weight loss, sarcopenia and adipopenia, fatigue, weakness, loss of appetite, and early satiety [42,60]. Tumorigenesis leads to cytokines release and eventually to metabolic pathways resulting in anorexia and hyper-catabolism. This means the hyper-catabolism of cancer cachexia is ascribed mainly to the systemic inflammatory response caused by advanced cancer. More recent works have suggested that IL-1, IL-6, IL-8, TNFα, and NF-κB are the most common pro-inflammatory cytokines in PDAC-dependent cachexia [12,60,61].

Metabolic alterations lead to the hyper-catabolism of muscle mass, which is one of the most common manifestations of cancer cachexia [12,26,60]. In addition, insulin resistance has also been suggested as a relevant mechanism involved in cancer cachexia. [12] Numerous interventional studies, such as those mentioned here, have validated that inflammation and reprogrammed metabolism may be more crucially involved in PDAC than previously assumed. Beyond the tissue level, the local tumor can affect host metabolism via cachexia, impairing antitumor immunity [26,42].

## 4. Microbiome-Induced Inflammation in PDAC

The microbiome’s influence on the development of many types of cancer has recently attracted increased attention in studies. The human microbiome is a major systemic homeostasis player. Most of them, mainly bacteria but also fungi and viruses, are known to deeply influence our physiology, from vitamin synthesis and the metabolism of carbohydrates, proteins, and fats to the modulation of the immune, endocrine and nervous systems (Table 2).

Over the past two decades, a large number of studies have discovered that microbiome-induced inflammation is associated with PDAC via multiple pathways, with several proinflammatory factors leading to impaired antitumor immune surveillance and altered cellular processes in TME, with the rapid development of genomics sequencing technology including next-generation sequencing, whole-genome shotgun metagenomics, 16S rRNA amplicon sequencing, and even 18S internal transcribed space sequencing, as well as novel algorithms in the field of computational science [62,63]. Furthermore, we discuss these microbial species’ potential clinical implications as prognostic and diagnostic biomarkers. Rebiotics, probiotics, antibiotics, fecal microbial transplantation, and bacteriophage therapy have all been shown in previous studies to have potentially favorable impacts on microbial diversity, which improves treatment outcomes.

### 4.1. Microbial Dysbiosis

#### 4.1.1. The Oral Microbiome, Periodontal Disease, and Tooth Loss

Some oral bacteria including *Porphyromonas gingivalis* (*P. gingivalis*), *Fusobacterium*, *Neisseria elongate,* and *Streptococcus mitis* have been shown to confer augmented susceptibility to PDAC using 16S rRNA sequencing. Likewise, periodontal disease and tooth loss are prospectively associated with an increased risk of PDAC [64]. Periodontitis is a chronic oral inflammation of the gingiva and surrounding tissues. It is the most common infectious condition leading to tooth loss and has been linked to various cancers of the pancreas [7,65]. In particular, *P. gingivalis*, a Gram-negative anaerobic pathogen, has been linked to a high risk of developing PDAC. Scientists have hypothesized that *P. gingivalis* plays a critical role in initiating inflammation and escaping the immune response related to lipopolysaccharide (LPS) and toll-like receptors (TLRs) [10,17,65,66]. More specifically, *Fusobacterium* could increase the production of ROS and inflammatory cytokines, modulate the tumor immune microenvironment, and drive myeloid cell infiltration in intestinal tumors.

#### 4.1.2. *Helicobacter pylori*

*Helicobacter pylori* is a well-known bacterium that colonizes the human stomach. Scientists have found that the *H. pylori* IgG level was higher in PDAC. Once human pancreatic cells are infected with *H. pylori*, they can colonize the pancreas and may be associated with the malignant potential of adenocarcinoma. A preclinical study put forward that direct *H. pylori* colonization in pancreatic cancer cells was associated with the activation of molecular pathways controlling PDAC growth and progression. Coincidentally, another research group discovered that *H. pylori* was found in the duodenum at a higher frequency in PDAC patients than in normal controls. Similarly, other studies used meta-analysis to confirm that *H. pylori* was associated with an increased risk of pancreatic cancer in humans. *H. pylori* may promote the development of PDAC by causing chronic mucosal inflammation as well as changes in cell proliferation and differentiation [7,62,65,67,68].

#### 4.1.3. The HBV and HCV

Hepatotropic viruses *HBV* and *HCV* can be found in extrahepatic tissues, including the pancreas, and may play a role in the development of extrahepatic malignancies. Specifically, HBsAg and HbcAg were found in the cytoplasm of pancreatic acinar cells. Inducing inflammation and modifying tissue viscoelasticity, DNA integration in infected cells that delay host immune system clearance of *HBV/HCV*-containing cells, and modulating the PI3K/AKT signaling pathway via the *HBV/HCV* protein are all possible mechanisms of the contribution of *HBV* or *HCV* to carcinogenesis [7,65].

#### 4.1.4. The Pancreatic Microbiome

In general, the pancreas was considered to be a sterile organ. Geller et al. [69] discovered *Gammaproteobacteria* in PDAC tissue specimens with gemcitabine resistance and hypothesized that this type of bacteria might modulate tumor sensitivity to gemcitabine. Pushalkar et al. [9] investigated the role of the intratumoral microbiome in PDAC progression and immunotherapy response modulation. Through a longitudinal analysis between age-matched KC (p48^Cre^; LSL-Kras^G12D^) and wild-type mice, certain bacterial populations were found to be enriched in KC mice, with the most abundant species being *Bifidobacterium pseudolongum*. Thus, the pancreas is not sterile and has its own microbial environment, which may affect the occurrence and development of PDAC. Halimi, A., et al. [11] discovered that *Gammaproteobacteria* and Bacilli dominated in cultivating the pancreatic microbiome using MALDI-TOF MS profiling analysis.

#### 4.1.5. Other: Fungi and Viruses 

Many new studies in mice and humans have demonstrated that fungi migrating from the gut to the pancreas might be a contributor to pathogenesis in PDAC. Aykut, B. and Pushalkar, S., et al. [70] discovered using DNA Sequencing that the ligation of mannose-binding lectin (MBL), which binds fungal wall glycans to activate the complement cascade, was required for oncogenic progression. Oncogenic Kras-induced inflammation leads to fungal dysbiosis, which in turn promotes tumor progression via activation of the MBL-C3 cascade. This research indicates a previously unknown MBL-C3-dependent role of the pancreatic mycobiome in PDAC, providing a basis for further work uncovering disease biomarkers or aiding in the development of new therapies [17,62,70]. Similarly, Alam et al. [71] revealed that the fungal microbiome drives a Kras^G12D^-MEK signaling pathway in PDAC cells that promotes the secretion of a pro-inflammatory cytokine, IL-33. Furthermore, IL-33 recruits and activates TH_2_ and innate lymphoid cells 2 (ILC2), which can subsequently hasten the spread of PDAC by secreting pro-tumorigenic cytokines such as IL-4, IL-5, and IL-13.

### 4.2. Microbial Metabolites 

Many researchers have devoted themselves to exploring how microbial metabolites trigger other chain reactions. Pushalkar et al. [9] reported that a cancerous pancreas exhibits a distinct microbial metabolite profile that is significantly more prevalent than that of a healthy pancreas in both murine models and humans. Notably, this includes lipopolysaccharide (LPS), a component of Gram-negative bacterial cell walls, which can engage multiple Toll-like receptor (TLR) signaling pathways, potentially resulting in T-cell anergy [9,65,68]. We also have the information from the review of Orlacchio, A. and P. Mazzone, which suggests that the activation of TLRs initiates a signaling cascade, which in turn leads to activation of the transcription factor involved in inflammation cytokines that are necessary to activate potent immune responses and promote tumorigenesis [66,72]. LPS-TLR signaling can activate NF-κB, MAPK, and the signal transducers and activators of the STAT3 signaling pathway and trigger mutation of the Kirsten rat sarcoma viral oncogene (KRAS), which can promote PDAC progression [9,12,65,68].

Cao, H., et al. [73] and Wang, S., et al. [74] provided evidence that deoxycholic acid (DCA) and short-chain fatty acids (SCFAs), produced by the intestinal bacteria *Akkermansia* or *Clostridium*, both promote intestinal tumorigenesis in conjunction with adenomatous polyposis coli (Apc) gene mutations [12]. Some researchers are shedding light on the potential DCA-mediated intestinal dysbiosis mechanisms of causing DNA damage, impairing the intestinal barrier function, modulating the inflammatory cytokines and chemokines expressed in the PDAC TME, and ultimately promoting intestinal carcinogenesis via the activation of Wnt signaling, which is important for cell proliferation during tumorigenesis [65,68,73]. Most importantly, DCA can activate the epidermal growth factor receptor (EGFR) and promote the release of its ligand, amphiregulin, which is identified as an oncogenic factor by the DCA-induced EGFR, MAPK, and STAT3 signaling pathways in PDAC tumorigenicity [74].

## 5. Conclusions

The relationship between tumors and inflammation is a well-known clinical finding. The work discussed above has significantly improved our understanding of inflammation in PDAC, particularly with a focus on immunity, the tumor microenvironment, metabolism, and microbiome risk factors. In this paper, we propose that inflammation, an evolutionarily conserved response to damage aimed at restoring tissue integrity following injury, may be involved in tumorigenesis. The prospective role of inflammation in oncogenesis and progression in PDAC is discussed in depth from the level of pathways, chemokines, cytokines, and the many cells responsible for creating them, including immune cells, inflammatory cells, and tumor cells.

Emerging evidence supports the implementation of these discoveries to improve current therapies for PDAC. We know the treatment landscape for PDAC has been evolving, particularly with the exploration of immunotherapies and anti-inflammatory drugs. Checkpoint inhibitors, such as anti-PD-1 (pembrolizumab) and anti-CTLA-4, block inhibitory signals that prevent T cells from attacking cancer cells. In terms of cancer vaccines, three main techniques, namely peptide vaccines, nucleotide vaccines, and dendritic cell vaccines, are being used in clinical trials to activate T cells specifically against PDAC. Researchers are working to adapt Chimeric antigen receptor (CAR) T-cell therapy for solid tumors like PDAC by targeting specific antigens such mesothelin. Furthermore, growing tumor-infiltrating lymphocytes from a patient’s tumor and reintroducing them may boost the immune response to PDAC. Combination therapies involving chemotherapy or radiation may improve their efficacy.

Ultimately, we are committed to exploring the dynamic and evolving interplay between inflammation and PDAC and improving the response rate in the immunotherapy of PDAC in the future.

## Figures and Tables

**Figure 1 ijms-25-10991-f001:**
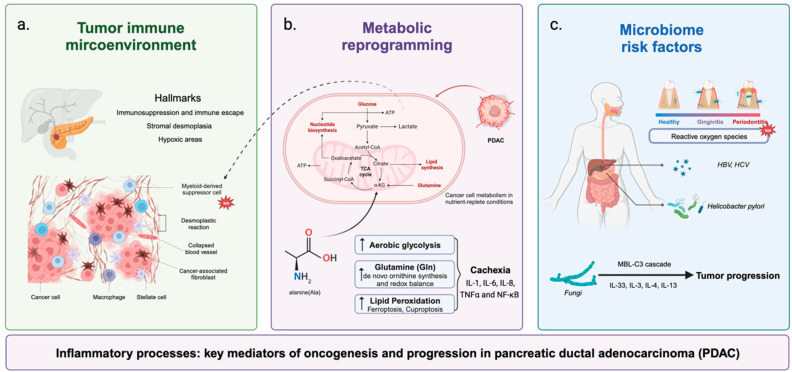
The dynamic interplay between inflammation and immune regulation, microenvironment alteration, metabolic reprogramming, and microbiome risk factors in pancreatic ductal adenocarcinoma. (**a**) Three hallmarks of PDAC tumor microenvironment; (**b**) inflammatory cells might affect and be affected by tumor cell metabolism; (**c**) microbiome-induced inflammation is associated with PDAC via multiple pathways with several proinflammatory factors.

**Figure 2 ijms-25-10991-f002:**
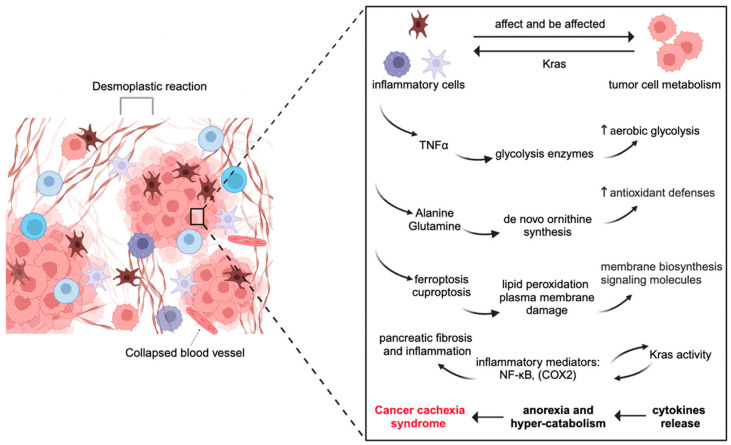
Inflammatory cells might affect and be affected by tumor cells from the energy metabolism of carbohydrates, proteins, and lipids.

**Table 1 ijms-25-10991-t001:** Inflammation and immune interactions in the PDAC tumor microenvironment.

Cell Types	Hallmarks of PDAC TME	Recruitment and Activation	Functions in PDAC
Regulatory T cells (Tregs)	Immunosuppression	IL-1α, NF-κB, TGF-β, TNFα and CCL5 derived from Cancer cell and inflammatory cells	Secretion of growth factors, IL-10, TGF-β, Forkhead box P3 (Foxp3), CTLA-4 and granzyme B to induce effector T cell cytolysis Tregs can bind to IL-2 competitively to inhibit effector cells.
Myeloid-derived suppressor cells (MDSCs)	Immunosuppression	Expansion under pathologic conditions including cancer and inflammation (Lactate ) Hypoxic feature of PDAC TME	MDSCs secrete IL-10, TGF-β, indoleamine 2,3-dioxygenase 1 (IDO1) and reactive oxygen species (ROS) to inhibit lymphocyte function and suppress T-cell responses IL-6 released by myeloid cells induces STAT3 activation to promote PanINs progression and PDAC development IL-1 and IL-17 also can facilitate tumor growth
Tumor-associated macrophages (TAMs)	Immunosuppression *Stromal desmoplasia*	TAMs mainly come from monocytes in peripheral blood	IL-6 induces STAT3 activation in the pancreas via IL-6 trans-signaling leading to disease progression. TAMs also can secrete IL-10, IL-17 and indoleamine 2,3-dioxygenase 1 (IDO1) to increase tumor growth indirectly. TAMs can stimulate angiogenesis to increase the tumor’s blood supply and promote metastasis by secreting VEGF and IL-35. CCL18 secreted by TAMs works by binding to PITPNM3 and induces VCAM-1 expression TGF-β1 released by TAMs upregulates PD-L1 levels in PDAC cells
Cancer-associated fibroblasts (CAFs)	Immunosuppression *Stromal desmoplasia*	IL-1β secreted from resident immune cells and the tumor cells can reprogram normal fibroblasts into pro-inflammatory CAFs	Activated CAFs further mediate tumor-enhancing inflammation by recruiting TAMs and promoting angiogenesis IL-6 can promote the differentiation of monocyte precursors into MDSCs via STAT3 and promote immune suppression CCL17, IL-15, and TGF-β also can promote Treg recruitment and differentiation CAFs can impair antitumor immunity through the production of soluble factors such as IL10, TGF-β, or VEGF
Pancreatic stellate cells (PSCs)	Immunosuppression *Stromal desmoplasia* Hypoxic areas	IL-1β, mitogenic and fibrogenic factors derived from tumor cell, regulated by pancreatic microbiome, promote the activation and secretory phenotype of quiescent PSCs Hypoxia can induce PSC activation	Soluble factors, including IL-6, VEGF-A, and stromal cell-derived factor-1 (SDF-1), favour angiogenic and inflammatory responses and invasion during PDAC progression.

**Table 2 ijms-25-10991-t002:** Microbiome-induced inflammation is associated with PDAC via multiple pathways with several proinflammatory factors.

Type	Microbiome	Functions in PDAC
Microbial dysbiosis	*P. gingivalis*	A critical role in initiating inflammation and escaping the immune response related to lipopolysaccharide (LPS) and toll-like receptors (TLRs)
	*Fusobacterium*	*Fusobacterium* could increase the production of ROS and inflammatory cytokines, and modulate the tumor immune microenvironment
	*Helicobacter pylori*	The *Helicobacter pylori* IgG level was higher in PDAC *Helicobacter pylori* may promote the development of PDAC by causing chronic mucosal inflammation as well as changes in cell proliferation and differentiation.
	Hepatotropic viruses *HBV* and *HCV*	HBsAg and HBcAg were found in the cytoplasm of pancreatic acinar cells Inflammation and modifying tissue viscoelasticity, DNA integration in infected cells that delay host immune system clearance of *HBV*/ *HCV*-containing cells Modulating the PI3K/AKT signaling pathway via the HBV/HCV protein
	*Gammaproteobacteria* *Bifidobacterium pseudolongum*	The pancreas is not sterile and has its own microbial environment which may affect the occurrence and development of PDAC. *Gammaproteobacteria* might modulate tumor sensitivity to gemcitabine.
	Other: Fungi and viruses	Oncogenic Kras-induced inflammation leads to fungal dysbiosis, which in turn promotes tumor progression via activation of the MBL-C3 cascade. The fungal microbiome drives a KrasG12D-MEK signaling pathway in PDAC cells that promotes the secretion of a pro-inflammatory cytokine, IL-33, which can subsequently hasten the spread of PDAC by secreting pro-tumorigenic cytokines such as IL-4, IL-5, and IL-13.
Microbial metabolites	lipopolysaccharide (LPS)	A Gram-negative bacterial cell wall component LPS can interact with several TLRs signaling pathways leading to T-cell anergy. LPS-TLR signaling can activate NF-κB, MAPK, the signal transducers and activators of STAT3 signaling pathway and trigger mutation of KRAS
	deoxycholic acid (DCA) and short-chain fatty acids (SCFAs)	They both promote intestinal tumorigenesis in conjunction with adenomatous polyposis coli (Apc) gene mutation Causing DNA damage, modulating the inflammatory cytokines and chemokines expressing in the PDAC TME via activation of Wnt signaling which is important for cell proliferation during tumorigenesis. DCA can activate the epidermal growth factor receptor (EGFR)

## Data Availability

No new data were created or analyzed in this study.

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
