# Peer review of "Inflammatory Processes: Key Mediators of Oncogenesis and Progression in Pancreatic Ductal Adenocarcinoma (PDAC)"

_ijms, 2024, doi:10.3390/ijms252010991_

Round 1

Reviewer 1 Report

Comments and Suggestions for Authors

This review provides a valuable overview of the role of inflammation in PDAC, but it would benefit from more detailed discussions of the microbiome, a stronger emphasis on the therapeutic potential of targeting inflammatory pathways, and a clearer critical analysis of current research gaps.

-        The abstract provides an informative summary of the review’s focus on the dynamic relationship between inflammation, immune regulation, metabolism, and the microbiome in PDAC progression. However, it could benefit from more specificity, particularly by outlining key findings or insights more clearly.

-        The background is not sufficient and has critical flaws. The readers may benefit from some more background and context. For example, a short introduction on the ongoing preclinical efforts to identify novel targets and to develop novel strategies to treat pancreatic cancer would provide some orientation to readers and set the scene. For this purpose, I suggest seeing the following papers and adding the references:

1.     Molecules, 2023, 28(18), 6450. Doi: 10.3390/molecules28186450

2. Cancers (Basel). 2023 Jun 30;15(13):3423. doi: 10.3390/cancers15133423.

3.     Mar Drugs. 2023 May 4;21(5):288. doi: 10.3390/md21050288.

-        Moreover, the introduction could benefit from a deeper exploration of why inflammation is such a crucial factor in PDAC compared to other cancer types.

-        The review provides detailed discussions of key inflammatory mediators (e.g., cytokines, chemokines) and their roles in immune suppression and PDAC progression. The section on immune cells (e.g., Tregs, MDSCs) is particularly strong, with clear explanations of how these cells contribute to tumor immune evasion. But I suggest to expand the microbiome section, focusing on recent advances and specific microbial species or pathways that have been implicated in PDAC. See the following reference:

1.      BMC Microbiol 24, 16 (2024). https://doi.org/10.1186/s12866-023-03166-4

-        The discussion effectively summarizes the main points regarding inflammation’s role in PDAC and suggests that targeting these pathways could offer new therapeutic avenues. The conclusion provides a satisfactory closure to the review. I suggest to expand the discussion on therapeutic implications, particularly in the context of immunotherapies or anti-inflammatory drugs.

-        Review the manuscript for minor grammatical issues or awkward phrasing.

Comments on the Quality of English Language

Review the manuscript for minor grammatical issues or awkward phrasing.

Author Response

Thank you very much for taking the time to review our manuscript. Please find the detailed responses below and the corresponding corrections in track changes in the re-submitted files.

Comment 1: [The abstract provides an informative summary of the review’s focus on the dynamic relationship between inflammation, immune regulation, metabolism, and the microbiome in PDAC progression. However, it could benefit from more specificity, particularly by outlining key findings or insights more clearly.]

Response 1: Thanks for pointing this out, We have updated the key findings for the therapeutic potential of the exploration of immunotherapies and anti-inflammatory drugs targeting inflammatory pathways. [This review elucidates the dynamic interplay between inflammation and immune regulation, microenvironment alteration, metabolic reprogramming, and microbiome risk factors in PDAC, committing to exploring a deeper understanding of the role of crucial inflammatory pathways and molecules for providing insights into therapeutic strategies.]

Comment 2: [The background is not sufficient and has critical flaws. The readers may benefit from some more background and context. For example, a short introduction on the ongoing preclinical efforts to identify novel targets and to develop novel strategies to treat pancreatic cancer would provide some orientation to readers and set the scene. ]

Response 2: Thanks for pointing this out and suggestion for related references, We have added relevant content. [In recent years, emerging evidence demonstrates that inflammation is strongly associated with pancreatic cancer, potentially representing key mediators. [3, 4] In one mechanism, inflammation promotes the survival and growth of cancer cells through the production of inflammatory mediators such as cytokines. In another way, inflammation is a mediator of immunosurveillance suppression. In the broadest sense, inflammatory cells might affect, and be affected by, tumor cell metabolism. Considering all above factors, new discoveries and therapeutic avenues for PDAC have arisen simultaneously, particularly in the context of immunotherapies or anti-inflammatory drugs. [20-22]

Next, we focus on the dynamic and evolving interplay between inflammation with immune regulation, microenvironment alteration, metabolic reprogramming, and microbiome risk factors for PDAC, targeting various immune cells, tumor cells, inflammatory cells, mutation, chemokines, and cytokines in PDAC (Figure 1). And we also summarize the current research on inflammatory pathways and molecules that are expected to become effective therapeutic strategies for PDAC.]

Comment 3: [the introduction could benefit from a deeper exploration of why inflammation is such a crucial factor in PDAC compared to other cancer types.]

Response 3: [Inflammation is particularly important in PDAC compared to other cancer types for various reasons: PDAC is frequently linked to chronic pancreatitis; PDAC is distinguished by a dense fibrotic stroma, known as a desmoplastic response, which is triggered by inflammatory cells and cytokines; Inflammatory processes can induce DNA damage and mutations, contributing to the genetic instability that is characteristic of PDAC; PDAC is renowned for its ability to escape the immune system. The inflammatory environment can promote the recruitment of immune suppressive cells, preventing effective anti-tumor immunity. These factors collectively contribute to the aggressive nature of PDAC, making it distinct from other cancer types.]

Comment 4: [I suggest to expand the microbiome section, focusing on recent advances and specific microbial species or pathways that have been implicated in PDAC. See the following reference: 1.      BMC Microbiol 24, 16 (2024). https://doi.org/10.1186/s12866-023-03166-4]

Response 4: [In the past two decades, a large number of studies have discovered that microbiome-induced inflammation is associated with PDAC via multiple pathways with several proinflammatory factors leading to impaired antitumor immune surveillance and altered cellular processes in TME, with the rapid development of genomics sequencing technology including next-generation sequencing, whole-genome shotgun metagenomics, 16S rRNA amplicon sequencing and even 18S internal transcribed space sequencing, as well as novel algorithms in the field of computational science. [62, 63] Furthermore, we discuss these microbial species' potential clinical implications as prognostic and diagnostic biomarkers. Rebiotics, probiotics, antibiotics, fecal microbial transplantation, and bacteriophage therapy have all been shown in studies to have potentially favorable impacts on microbial diversity, which improves treatment outcomes.]

Comment 5: [The conclusion provides a satisfactory closure to the review. I suggest to expand the discussion on therapeutic implications, particularly in the context of immunotherapies or anti-inflammatory drugs.]

Response 5: [Emerging evidence supports the implementation of these discoveries to improve current therapies for PDAC. We know the treatment landscape for PDAC has been evolving, particularly with the exploration of immunotherapies and anti-inflammatory drugs. Checkpoint Inhibitors, such as anti-PD-1 (pembrolizumab) and anti-CTLA-4, block inhibitory signals that prevent T cells from attacking cancer cells. In terms of cancer vaccines, three main techniques, peptide vaccines, nucleotide vaccines, and dendritic cell vaccines, are being used in clinical trials to activate T cells specifically against PDAC. Researchers are working to adapt Chimeric antigen receptor (CAR) T-cell therapy for solid tumors like PDAC by targeting specific antigens such mesothelin. Furthermore, growing tumor-infiltrating lymphocytes from a patient's tumor and reintroducing them may boost the immune response to PDAC. Combination therapies involving chemotherapy or radiation may improve their efficacy.]

Thanks a lot.

Reviewer 2 Report

Comments and Suggestions for Authors

The paper "Inflammatory processes: key mediators of oncogenesis and progression in pancreatic ductal adenocarcinoma (PDAC)," explores how inflammation contributes to the development and progression of PDAC. It covers the interaction between inflammation, immune regulation, the tumor microenvironment, metabolic changes, and microbiome risk factors. The review highlights the roles of key immune cells like Tregs, MDSCs, and TAMs, as well as how inflammation-driven metabolic changes promote tumor survival. It gives a broad overview of current knowledge and suggests that targeting inflammation could improve immunotherapy for PDAC.

Major points:

The paper is well-organized, using up-to-date references and providing clear sections on immune cells, metabolism, and the microbiome, making it easy to follow.

However, it mostly repeats existing knowledge without discussing the current clinical impact. The review mentions potential therapeutic targets but does not explore them in enough detail. It lists information without drawing conclusions or explaining how this knowledge could be applied clinically. Adding a table of ongoing clinical trials and more concrete conclusions would strengthen the paper. The conclusion is weak and does not highlight the most relevant findings or future directions.

Minor points:

The writing also needs improvement. Some sentences are awkward or unclear, like: “Pushalkar, S., et al. [9] suggested that the malignant pancreas has a quite microbial metabolite that is significantly more widespread than the healthy pancreas in both mice and humans, such as LPS, a Gram-negative bacterial cell wall component, which can interact with several TLRs signaling pathways leading to T-cell anergy.” The language should be revised to make the meaning clearer.

Additionally, the capitalization of "tumor Microenvironment" on page 11, line 448, should be corrected.

In summary, the paper should be accepted with major revisions. 

These revisions should focus on adding clearer conclusions and a stronger discussion of clinical applications. The language also needs to be revised. 

Comments on the Quality of English Language

See my comments above: The writing also needs improvement. Some sentences are awkward or unclear, like: “Pushalkar, S., et al. [9] suggested that the malignant pancreas has a quite microbial metabolite that is significantly more widespread than the healthy pancreas in both mice and humans, such as LPS, a Gram-negative bacterial cell wall component, which can interact with several TLRs signaling pathways leading to T-cell anergy.” The language should be revised to make the meaning clearer.

Author Response

Thank you very much for taking the time to review our manuscript. Please find the detailed responses below and the corresponding corrections in track changes in the re-submitted files.

Comments 1: [The paper is well-organized, using up-to-date references and providing clear sections on immune cells, metabolism, and the microbiome, making it easy to follow.

However, it mostly repeats existing knowledge without discussing the current clinical impact. The review mentions potential therapeutic targets but does not explore them in enough detail. It lists information without drawing conclusions or explaining how this knowledge could be applied clinically. Adding a table of ongoing clinical trials and more concrete conclusions would strengthen the paper. The conclusion is weak and does not highlight the most relevant findings or future directions.]

Responses 1: Thanks for pointing therapeutic strategies out. We have summarized this part, from research to application, to make the article more complete and valuable.  [Please see the attachment.]

Comments 2: [The writing also needs improvement.] 

Responses 2: Thanks for pointing this out. We have made some changes to the writing aspect.

Thanks a lot.

Round 2

Reviewer 2 Report

Comments and Suggestions for Authors

The authors have sufficiently enhanced the manuscript.